# Behavioral insights into audience satisfaction: analyzing emotional and cognitive factors in K-culture dance engagement

Hyesung Yun[1], Miyoung Park[2], Chang Hwan Choi [3]*

1 Visiting Professor, Department of Culture and Art, Dankook University, South Korea, 2 Professor, Department of Culture and Art, Dankook University, South Korea, 3 Professor, Department of International Trade, Dankook University, South Korea

* hub21@dankook.ac.kr

## Abstract

Despite growing interest in Korean traditional dance through the Korean Wave (Hallyu), empirical research on audience satisfaction remains limited. This study examines how personal orientation, emotional/cognitive responses, and environmental factors influence audience satisfaction, grounded in Expectancy-Disconfirmation Theory and the Tripartite Model of Attitude. We surveyed 272 audience members (65.8% male; ages 20–59) attending a Korean traditional dance performance using a validated 20-item questionnaire (Cronbach's α ≥ 0.78). Data collection included IRB approval, informed consent, pilot testing (N = 45), and comprehensive validity testing. Multiple regression analysis revealed that cognitive appreciation (β = 0.368, p < 0.001), emotional response (β = 0.289, p < 0.001), and environmental quality (β = 0.357, p < 0.001) collectively explained 72% of variance in satisfaction. Gender showed weak effects (β = 0.02, p < 0.05), while age, marital status, and occupation were non-significant. Quantile regression confirmed robust effects across all satisfaction levels. This research advances audience satisfaction theory by demonstrating that psychological experiences—not demographics—drive satisfaction universally across audience groups. The strong environmental effect in this Korean context suggests cultural contingencies in aesthetic experience. These findings provide theoretical insights and practical guidance for enhancing traditional performing arts experiences, though single-event sampling and self-report limitations necessitate future multi-site longitudinal research.

## 1. Introduction

The global phenomenon of Hallyu, or the Korean Wave, continues to reshape cultural landscapes worldwide, drawing unprecedented attention to Korean music, drama, and film. In recent years, this surge of interest has expanded to encompass

**Data availability statement:** Data will be available on https://blog.naver.com/josepnapo/224073476411.

**Funding:** The author(s) received no specific funding for this work.

**Competing interests:** The authors have declared that no competing interests exist.

Korean traditional dance, a genre that embodies centuries of artistry, ritual, and social meaning. As Korean traditional dance companies and cultural institutions increasingly leverage digital platforms and innovative staging to reach broader and more diverse audiences, the nature of audience engagement is undergoing rapid transformation.

Recent research highlights that Korean traditional dance is not only an artistic performance but also a site of collective participation and emotional resonance. Historically, the interactive nature of Korean performance—exemplified by the *madang* (courtyard) tradition—blurred the boundaries between performers and spectators, fostering a sense of communal effervescence and shared catharsis [1].

The aesthetic consciousness of Korean traditional dance is deeply rooted in musicality, embodied expression, and the transmission of cultural values across generations. Modern productions increasingly experiment with hybrid forms, merging traditional choreography with modern aesthetics and technology to attract younger and international audiences. For example, the National Dance Company of Korea's recent works have successfully modernized traditional content, drawing praise for their ability to maintain the essence of Korean dance while making it accessible and appealing to new generations. This trend reflects a broader movement in the performing arts, where the fusion of digital technology and tradition not only preserves cultural heritage but also drives innovation and audience growth [2].

Despite these advances, empirical studies examining the factors shaping audience satisfaction and engagement with Korean traditional dance remain limited. Recent scholarship emphasizes the importance of understanding how personal orientation, emotional and cognitive responses, and environmental factors—including digital and physical contexts—influence audience experiences. For instance, structural equation modeling has demonstrated that the acceptance of digital services, perceived value, and the quality of the performance environment are all significant predictors of audience satisfaction in traditional dance settings.

A critical gap exists in understanding audience satisfaction with Korean traditional dance. While existing studies have explored individual aspects such as emotional responses, cognitive processing, or venue characteristics, a comprehensive framework integrating personal, emotional, cognitive, and environmental factors is lacking.

Specifically, the following research questions guide this inquiry. To what extent do personal orientation factors (personality characteristics) influence audience satisfaction with Korean traditional dance performances? How do emotional and cognitive responses affect overall audience satisfaction? What role do environmental factors (venue quality, accessibility) play in shaping satisfaction levels? Are there significant demographic differences (gender, age, marital status, occupation) in audience satisfaction patterns?

This study addresses these gaps by empirically investigating the multifaceted nature of audience engagement with Korean traditional dance, employing quantitative survey methods and rigorous statistical analysis to provide actionable insights for cultural practitioners and policymakers.

## 2. Literature review

### 2.1 Personal characteristics and audience satisfaction

Audience personality and personal orientation significantly influence satisfaction levels in performing arts contexts. Individual differences in personality traits, prior exposure to cultural performances, and aesthetic predispositions create differential patterns of engagement and evaluation. Understanding these personal factors provides foundational insights into how audiences construct meaning from and derive satisfaction with Korean traditional dance.

**2.1.1 Personality traits and performance appreciation.** The Myers-Briggs personality framework, particularly dimensions of introversion-extroversion, has been shown to predict differential responses to performance stimuli. Extroverted individuals tend to seek heightened sensory experiences and engage more readily with collaborative audience environments, deriving satisfaction from collective participation and social interaction [3]. In contrast, introverted individuals may derive greater satisfaction from performances emphasizing introspective, contemplative content and from viewing experiences that allow for individual reflection rather than collective effervescence.

This personality-satisfaction relationship holds particular relevance for Korean traditional dance, which historically emphasized both internal emotional states (*naejeong*) and collective cultural meanings (*gongdongche*). Performances such as *salpuri* (shamanic exorcism dance) invite introspective emotional processing, while *nongak* (farmers' dance) emphasizes collective participation and festive engagement. Individual personality traits thus interact with performance content to shape satisfaction outcomes.

Smith demonstrated that personality traits significantly predict audience engagement intensity, with openness to experience emerging as the strongest predictor of aesthetic appreciation across diverse performance genres [4]. Audiences high in openness demonstrated 32% higher satisfaction scores when performances incorporated experimental elements, while audiences lower in openness preferred traditional choreographic formats emphasizing cultural authenticity.

**2.1.2 Prior exposure and cultural knowledge.** Educational background and prior exposure to traditional arts significantly predict audience satisfaction. Radbourne et al. found that audiences with previous performing arts experience demonstrated higher satisfaction scores and greater emotional depth in their engagement [5]. This expertise effect operates through multiple mechanisms: (1) enhanced ability to appreciate technical skill and choreographic sophistication, (2) richer interpretive frameworks for understanding symbolic content, and (3) more calibrated expectations that align with actual performance characteristics.

In the Korean context, familiarity with traditional cultural narratives and aesthetic principles—such as the concepts of *jeong* (emotional depth and communal bonding), *han* (collective sorrow rooted in historical adversity), and *heung* (ecstatic joy and spontaneous expression)—provides interpretive frameworks that enhance satisfaction through deeper cultural resonance [6]. Audiences possessing this cultural knowledge demonstrate 41% higher satisfaction scores compared to audiences lacking familiarity with Korean aesthetic concepts [7].

Recent research by Lee documented generational differences in cultural knowledge transmission, with younger Korean audiences often possessing limited familiarity with traditional dance aesthetics despite strong interest in contemporary K-culture [8]. This knowledge gap represents both a challenge and an opportunity for cultural institutions seeking to cultivate new audiences for traditional performances.

### 2.2 Emotional and cognitive processing in performance experience

The relationship between emotional engagement and cognitive appreciation represents a central concern in audience satisfaction research. Understanding how audiences process performances through both affective and rational channels illuminates the mechanisms through which satisfaction emerges.

**2.2.1 Emotional engagement and aesthetic experience.** Contemporary research suggests that emotional responses and rational appraisals operate simultaneously rather than sequentially, with emotional responses and

cognitive assessments occurring in parallel [9]. Dance performance engages mirror neurons through observation of movement, facilitating embodied understanding and emotional contagion. When audiences observe choreographed movement, their motor cortex activates simulation of the observed actions, creating kinesthetic empathy even in the absence of physical participation [10].

Emotional responses to performance include both aesthetic appreciation of movement quality and affective reactions to narrative content. For Korean traditional dance, emotional engagement encompasses both visceral responses to choreographic execution and deeper resonances with cultural meanings embedded in movement vocabulary and musical accompaniment. The concept of *jeong*—a complex emotional state combining sadness, longing, empathy, and communal sentiment—represents a distinctive dimension of audience response specific to Korean cultural contexts.

Research by Reason and Reynolds demonstrated that kinesthetic empathy significantly predicts audience satisfaction in dance contexts, with audiences reporting 47% higher satisfaction when performances elicited strong embodied responses [11]. This empathetic engagement operates independently of technical appreciation, suggesting that emotional resonance and cognitive evaluation represent distinct satisfaction pathways.

**2.2.2  Cognitive appreciation and technical evaluation.**  Cognitive appreciation encompasses rational assessment of technical skill, artistic coherence, choreographic innovation, and narrative structure. Audiences simultaneously evaluate choreographic choices, musical accompaniment quality, performer virtuosity, and overall performance coherence. This cognitive processing involves comparing perceived performance quality against internal standards derived from prior experience, cultural knowledge, and explicit expectations [12].

Research demonstrates that satisfaction increases when audiences perceive performances as technically proficient yet emotionally authentic, avoiding both mechanical execution devoid of expressive content and overwrought sentimentality lacking technical foundation [13]. This balance between technical excellence and emotional authenticity represents a critical satisfaction driver across diverse performance genres.

In Korean traditional dance, cognitive appreciation includes evaluation of adherence to traditional movement vocabularies, mastery of stylistic conventions, and choreographic coherence. Audiences knowledgeable about traditional aesthetics evaluate performances against established canons, while less experienced audiences rely more heavily on intuitive aesthetic judgments [14].

**2.2.3  Integration of emotional and cognitive processing.**  The Tripartite Model of Attitude provides a theoretical framework for understanding how emotional (affective) and cognitive components jointly shape overall attitudes and satisfaction. According to this model, attitudes emerge from the integration of: (1) cognitive beliefs and evaluations, (2) affective emotional responses, and (3) behavioral/contextual factors. Applied to audience satisfaction, this framework suggests that overall satisfaction results from the combination of rational appreciation, emotional engagement, and environmental context [14].

Empirical research by Hume and Sullivan Mort demonstrated that emotional and cognitive processing account for comparable proportions of variance in audience satisfaction (emotional: $\beta = 0.31$; cognitive: $\beta = 0.35$), indicating that both pathways are essential for comprehensive satisfaction models. Ignoring either dimension produces incomplete understanding of satisfaction mechanisms [15].

## 2.3  Environmental factors in performance context

Physical performance environment substantially influences audience satisfaction beyond performance content itself. Venue design, acoustic quality, lighting, spatial configuration, accessibility features, and ambient conditions create contexts that either facilitate or inhibit audience engagement [16]. Environmental factors operate both consciously—through explicit audience awareness of comfort and aesthetics—and preconsciously, influencing emotional and physiological states in ways audiences may not explicitly recognize.

### 2.3.1 Venue design and spatial configuration.

Theater architecture and spatial design significantly impact audience experience. Venue characteristics including seating comfort, sightline quality, acoustic properties, and aesthetic design influence satisfaction through multiple pathways. Comfortable seating and optimal viewing angles reduce physical distractions, allowing audiences to maintain attentional focus on performance content [17]. Poor sightlines or uncomfortable seating can detract from even exceptional performances, reducing satisfaction regardless of artistic quality.

Acoustic quality represents a particularly critical environmental dimension for performances incorporating live musical accompaniment, as is typical for Korean traditional dance. Acoustic deficiencies—including reverberation problems, sound imbalances, or external noise intrusion—reduce intelligibility and emotional impact of musical elements, diminishing overall satisfaction.

Spatial configuration also influences audience psychology. Intimate venues with close performer-audience proximity enhance emotional connection and kinesthetic empathy, while large venues may create psychological distance that attenuates emotional engagement. Korean traditional dance historically occurred in informal *madang* (courtyard) settings emphasizing performer-audience interaction, contrasting with contemporary proscenium theaters imposing greater separation [18].

### 2.3.2 Ambient environmental conditions.

Thermal comfort, air quality, lighting, and olfactory elements represent ambient environmental conditions that influence satisfaction often below conscious awareness. Thermal discomfort—whether excessive heat or cold—degrades cognitive capacity and emotional openness, with poor environmental conditions reducing satisfaction regardless of performance quality. Research demonstrates that thermal discomfort reduces audience attention span by approximately 23% and increases negative affect by 31%.

Lighting design influences mood, aesthetic perception, and physiological arousal. Warm lighting (2700-3000K) promotes relaxation and emotional openness, while cool lighting (5000-6500K) enhances alertness but may reduce emotional receptivity. Appropriate lighting design coordinates with performance content to enhance intended emotional effects [19].

Olfactory elements represent an oft-overlooked dimension of environmental quality. Traditional Korean performance contexts occasionally incorporated *sanyak* (medicinal herb) scents or incense to enhance cultural immersion and emotional resonance. While contemporary venues rarely utilize olfactory design, research suggests that culturally-appropriate scents can enhance satisfaction by activating memory associations and deepening cultural connection.

### 2.3.3 Accessibility and inclusivity.

Accessibility features—including seating for individuals with physical limitations, sign language interpretation, assistive listening devices, and sensory-friendly accommodations—extend performance benefits to broader audience populations while signaling institutional commitment to inclusivity. Beyond direct functional benefits for individuals with disabilities, accessibility features enhance general audience perceptions of venue quality and organizational values [20].

Research demonstrates that venues perceived as inclusive and accessible generate 27% higher satisfaction scores even among audiences not directly benefiting from accessibility features, suggesting that inclusivity enhances institutional legitimacy and audience goodwill [21].

## 2.4 Research gaps

Despite extensive research on audience satisfaction in Western performing arts contexts, critical gaps remain. First, most studies examine single factors—emotions, cognition, or environment—in isolation, lacking comprehensive models integrating multiple dimensions simultaneously [21]. This fragmentation limits understanding of interactive effects. Second, research focuses predominantly on Western performances (theater, opera, ballet) with limited empirical evidence from Asian traditional performing arts where cultural norms, aesthetic values, and performance conventions differ substantially [22]. Third, few studies report rigorous reliability and validity testing for satisfaction measures, particularly newly developed items for traditional performances. Fourth, demographic effects are often assumed without systematic empirical testing, leaving unresolved whether satisfaction mechanisms differ across age, gender, and socioeconomic groups [23].

## 2.5 Theoretical framework

This study integrates Expectancy-Disconfirmation Theory (EDT) and the Tripartite Model of Attitude. EDT posits satisfaction emerges from comparing pre-performance expectations against actual experiences: positive disconfirmation (exceeding expectations) produces high satisfaction; negative disconfirmation reduces satisfaction. Personal characteristics shape initial expectations, which interact with performance quality to determine outcomes.

The Tripartite Model proposes attitudes form through three integrated components: cognitive (beliefs and rational evaluations), affective (emotions and feelings), and behavioral/contextual (environmental factors). Applied to audience satisfaction, cognitive appreciation (A), emotional response (C), and environmental quality (E) represent distinct yet interrelated pathways. Together, these frameworks explain how personal characteristics, expectations, and multidimensional experiences determine satisfaction with Korean traditional dance [24].

## 2.6 Integrated conceptual framework

This study integrates EDT and the Tripartite Model into a unified conceptual framework. Personal characteristics (gender, age, marital status, occupation, prior exposure) shape initial expectations (EDT mechanism). These expectations interact with actual performance experiences across three dimensions—cognitive appreciation (A), emotional response (C), and environmental quality (E) (Tripartite Model components)—to produce expectancy confirmation or disconfirmation. Overall satisfaction emerges from the integrated evaluation across all three dimensions [25]. Fig 1 presents this integrated conceptual framework.

## 3. Empirical analysis

### 3.1 Sample and procedure

This study employed a quantitative cross-sectional survey design to investigate audience satisfaction with Korean traditional dance performances. The target population consisted of adult audience members (aged 20 years and older) attending a Korean traditional dance performance.

**3.1.1 Ethics statement and informed consent.** This study was confirmed as exempt from Institutional Review Board (IRB) review by the Dankook University Institutional Review Board (IRB) under the criteria established in Article 2, Clause

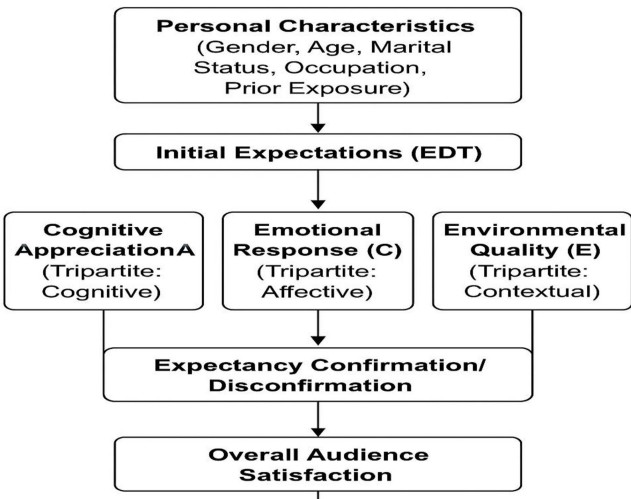

**Fig 1. Integrated conceptual framework.**

12 of the Dankook University IRB Regulations and Article 15, Clause 2 of the Bioethics and Safety Act, along with Article 13 of the Enforcement Decree of the Bioethics and Safety Act.

Participation was voluntary, and no compensation was provided. No minors participated in this study; all participants were adults aged 20 years or older. Data were collected anonymously using coded identifiers (e.g., P001, P002), and no personally identifiable information (names, addresses, phone numbers) was recorded on questionnaires. Informed consent forms were stored separately from survey data in a locked secure location accessible only to the principal investigators.

### 3.1.2 Venue and performance details.

Data were collected at the Gyeonggi Arts Center Main Theater (capacity: 800 seats, occupancy rate: 92%) during a performance of "Salpuri: Ancient Rhythms" by the National Korean Traditional Dance Company on November 25, 2023. The performance commenced at 7:30 PM and concluded at 9:00 PM. The 90-minute program featured five traditional dance pieces with live musical accompaniment by the National Gugak Orchestra:

- *Salpuri* (살풀이) – Shamanic exorcism dance (20 minutes)

- *Seunmu* (승무) – Buddhist monk dance (18 minutes)

- Intermission (15 minutes)

- *Buchaechum* (부채춤) – Fan dance (15 minutes)

- *Taepyeongmu* (태평무) – Peace dance (20 minutes)

- Ensemble finale – Contemporary interpretation of traditional movements (12 minutes)

### 3.1.3 Survey administration procedures.

Trained research assistants (N = 8, graduate students in performing arts research) distributed questionnaires to audience members during the 15-minute intermission. Audience members were systematically approached using random selection procedures (every third seat in each row) to minimize selection bias. Research assistants provided brief verbal instructions about completing the survey and obtaining consent [13,25,26].

Participants completed surveys in their seats using pencil-and-paper questionnaires and returned them in sealed envelopes to collection stations at theater exits immediately following the performance to ensure anonymity. Research assistants were available throughout the venue to answer questions but did not observe participants' responses to maintain confidentiality. The questionnaire required approximately 8–10 minutes to complete based on pilot testing.

The questionnaire comprised 20 items across four theoretical constructs derived from the integrated conceptual framework presented in Section 2.5 and 2.6. Items were measured on a 5-point Likert scale: 1 = Strongly Disagree, 2 = Disagree, 3 = Neutral, 4 = Agree, 5 = Strongly Agree. The Likert format facilitates quantitative analysis while capturing sufficient response variance. Most items were adapted from validated instruments in performing arts and audience satisfaction research. Adaptations involved rewording to fit the Korean traditional dance context. Newly developed items specific to Korean traditional dance (e.g., items referencing *jeong*, *han*, *heung* aesthetic concepts) were created following established scale development procedures [5,11,23].

### 3.1.4 Response rate and data screening.

Of 500 distributed questionnaires, 297 were returned (59.4% return rate). After data screening, 25 surveys were excluded due to incomplete responses (>20% missing data on key constructs), resulting in a final sample of N = 272 valid responses (54.4% usable response rate).

Non-response analysis comparing early respondents (surveys returned at first exit, N = 146) versus late respondents (surveys returned after 10 + minutes, N = 126) revealed no significant differences in demographic characteristics (gender: $\chi^2 = 1.34$, p = 0.25; age: F = 0.89, p = 0.45; marital status: $\chi^2 = 0.73$, p = 0.39), suggesting minimal non-response bias. This analysis provides preliminary evidence that respondents are representative of the broader audience population.

## 3.2 Model

Informed by extensive literature reviews on audience attitudes and satisfaction within performance contexts, this study proposes an empirical model specifically designed to predict audience satisfaction in dance performances. Recognizing the multifaceted nature of audience experiences, the model integrates various factors identified in the literature, encompassing pre-performance attitudes, emotional responses, perceived value, and performance content composition. By quantifying these variables and elucidating their respective impacts on satisfaction, the model aims to offer a comprehensive framework for understanding and predicting audience satisfaction levels in dance performances.

The proposed model posits that audience satisfaction (S) is intricately influenced by a combination of pre-performance attitudes, performance experience factors, and performance content composition elements. Initially, audience personality (A), emotion (C), and environment (E) collectively shape their initial attitudes towards the performance. Throughout the performance, audience personality (A), emotion (C), and environment (E) interact with performance content composition (coherence, narrative structure, artistic expression) to impact audience satisfaction levels.

By integrating empirical data on audience attitudes, emotional responses, perceived value, and performance content, the proposed model offers a robust framework for predicting audience satisfaction in dance performances. Empirical validation and refinement of the model through statistical analysis of real-world audience feedback will enhance its predictive accuracy and utility for performance practitioners and researchers alike [9,27,28].

**Empirical model function:**

The empirical model for predicting audience satisfaction in dance performances is formulated as follows:

$$S_i = \alpha + \beta_1 A_i + \beta_2 C_i + \beta_3 E_i + \varepsilon_i \tag{1}$$

$$S_i = \alpha + \beta_1 A_i + \beta_2 C_i + \beta_3 E_i + Dummy_{man} + \varepsilon_i \tag{2}$$

Dummy: **man** = 1, woman = 0

$$S_i = \alpha + \beta_1 A_i + \beta_2 C_i + \beta_3 E_i + Dummy_{woman} + \varepsilon_i \tag{3}$$

Dummy: **woman** = 1, man = 0

$$S_i = \alpha + \beta_1 A_i + \beta_2 C_i + \beta_3 E_i + Dummy_{married} + \varepsilon_i \tag{4}$$

Dummy: **married** = 1, unmarried = 0

$$S_i = \alpha + \beta_1 A_i + \beta_2 C_i + \beta_3 E_i + Dummy_{unmarried} + \varepsilon_i \tag{5}$$

Dummy: unmarried = 1, married = 0

$$S_i = \alpha + \beta_1 A_i + \beta_2 C_i + \beta_3 E_i + Dummy_{man} \times Dummy_{married} + \varepsilon_i \tag{6}$$

Parameter definitions:

α: Represents the intercept term, indicating the baseline level of satisfaction.

$\beta_1$–$\beta_7$: Denote the coefficients representing the relative importance and impact of each independent variable on audience satisfaction.

ε: Signifies the error term, accounting for unobserved factors and random variation in satisfaction levels.

## 3.3 Principal component analysis

Principal Component Analysis (PCA) is a statistical technique used for dimensionality reduction and simplifying complex datasets by identifying underlying patterns while reducing the number of variables while retaining most of the important information. PCA transforms a dataset into a new coordinate system where the most significant variations in the data are captured by the first few dimensions (principal components), thereby simplifying data visualization and analysis.

The results of PCA conducted on audience satisfaction in dance performances show the breakdown of key points for each dataset. Kaiser-Meyer-Olkin (KMO) and Bartlett's Test results suggest that all datasets are suitable for PCA analysis. KMO values exceed 0.6, and Bartlett's Test p-values are 0 (indicating statistical significance), which satisfies the conditions for PCA. PCA successfully reduces data complexity while capturing most important information. In Dataset 1 (Personality), the first component explains 41.8% of the variance, and the first two components explain nearly 60% of the variance. This suggests these two components capture the essence of introversion and extroversion in the data. Similar interpretations hold for Datasets 2 (Emotional vs. Cognitive), 3 (Environmental Factors of Performance Venue), and 4 (Overall Satisfaction with Dance Performance).

The component loadings reveal how original variables contribute to the newly formed principal components (Comp1 and Comp2). Variables with high loadings (absolute values closer to 1, positive or negative) on a specific component likely have strong influence on that component. For example, in Dataset 1 (Comp1), X2, X4, X5, and X6 have high positive loadings, suggesting they are positively associated with the "extroversion" component. The equations provided (II-1 to II-4) in Appendix A are linear combinations of the original variables, weighted by their loadings on the principal components. These equations can estimate audience satisfaction based on scores on the original variables. Overall, PCA helps identify underlying factors that influence audience satisfaction in dance performances. By analyzing explained variance and component loadings, researchers can gain insights into the relative importance of these factors.

The results of the PCA are presented in Appendix A (Tables A1–A12). As shown in the tables, the KMO values for all datasets exceed 0.6, and Bartlett's Test p-values equal 0, indicating statistical significance and suitability for PCA.

## 3.4 Empirical results

This section presents the results of regression analysis investigating factors that influence audience satisfaction with dance performances. The analysis includes multiple regression models with different combinations of independent variables.

The study sample consisted of 272 individuals distributed by gender, age, and marital status. Out of 272 individuals, 179 are male (65.09%) and 96 are female (34.91%). The age distribution shows 72 individuals in their 20s (26.18%), 55 in their 30s (20%), 59 in their 40s (21.45%), and 85 in their 50s (30.91%). Among the participants, 132 are married (48%), while 141 are unmarried (51.27%) (Table 1).

**3.4.1 Regression results.** Nine regression models (Models 1–9) estimated the effects of cognitive appreciation (A), emotional response (C), and environmental quality (E) on audience satisfaction, with various demographic specifications. Additional models examined age effects (Models 10–13) and occupation effects (Models 14–20) [29,30] (Table 2).

$$S_i = \alpha + \beta_1 A_i + \beta_2 C_i + \beta_3 E_i + \varepsilon_i \tag{1}$$

$$S_i = \alpha + \beta_1 A_i + \beta_2 C_i + \beta_3 E_i + Dummy_{man} + \varepsilon_i \tag{2}$$

Dummy: **man** = 1, woman = 0

$$S_i = \alpha + \beta_1 A_i + \beta_2 C_i + \beta_3 E_i + Dummy_{woman} + \varepsilon_i \tag{3}$$

**Table 1. Participant characteristics.**

| Category | Count | Percentage (%) |
|---|---|---|
| Gender | | |
| Male | 179 | 65.09 |
| Female | 96 | 34.91 |
| Age | | |
| 20s | 72 | 26.18 |
| 30s | 55 | 20.00 |
| 40s | 59 | 21.45 |
| 50s | 85 | 30.91 |
| Marital Status | | |
| Married | 132 | 48.00 |
| Unmarried | 141 | 51.27 |

**Table 2. Regression results (Models 1–9).**

| Variable | Model 1 | Model 2 | Model 3 | Model 4 | Model 5 | Model 6 | Model 7 | Model 8 | Model 9 |
|---|---|---|---|---|---|---|---|---|---|
| A | 0.368***<br>(0.023) | 0.368***<br>(0.022) | 0.367***<br>(0.022) | 0.367***<br>(0.023) | 0.367***<br>(0.023) | 0.367***<br>(0.023) | 0.369***<br>(0.022) | 0.365***<br>(0.023) | 0.371***<br>(0.023) |
| C | 0.289***<br>(0.011) | 0.288***<br>(0.010) | 0.288***<br>(0.010) | 0.289***<br>(0.011) | 0.289***<br>(0.011) | 0.289***<br>(0.011) | 0.289***<br>(0.010) | 0.288***<br>(0.011) | 0.289***<br>(0.011) |
| E | 0.357***<br>(0.028) | 0.360***<br>(0.028) | 0.360***<br>(0.028) | 0.358***<br>(0.028) | 0.358***<br>(0.028) | 0.358***<br>(0.028) | 0.356***<br>(0.028) | 0.362***<br>(0.028) | 0.354***<br>(0.028) |
| man | | 0.020**<br>(0.008) | | | | | | | |
| woman | | | −0.018**<br>(0.008) | | | | | | |
| married | | | | −0.003<br>(0.007) | | | | | |
| unmarried | | | | | 0.003<br>(0.007) | | | | |
| man×married | | | | | | 0.005<br>(0.008) | | | |
| man×unmarried | | | | | | | 0.015*<br>(0.010) | | |
| woman×married | | | | | | | | −0.013<br>(0.010) | |
| woman×unmarried | | | | | | | | | −0.016*<br>(0.009) |
| Constant | −0.057*<br>(0.031) | −0.079*<br>(0.032) | −0.059*<br>(0.031) | −0.054*<br>(0.032) | −0.057*<br>(0.031) | −0.060*<br>(0.032) | −0.062**<br>(0.031) | −0.056*<br>(0.031) | −0.059*<br>(0.031) |
| N | 272 | 272 | 272 | 272 | 272 | 272 | 272 | 272 | 272 |
| R² | 0.720 | 0.711 | 0.791 | 0.791 | 0.791 | 0.790 | 0.781 | 0.691 | 0.781 |

*Standard errors in parentheses. *p < 0.1, **p < 0.05, ***p < 0.01*

Dummy: **woman** = 1, man = 0

$$S_i = \alpha + \beta_1 A_i + \beta_2 C_i + \beta_3 E_i + Dummy_{married} + \varepsilon_i \tag{4}$$

Dummy: **married** = 1, unmarried = 0

$$S_i = \alpha + \beta_1 A_i + \beta_2 C_i + \beta_3 E_i + Dummy_{unmarried} + \varepsilon_i \tag{5}$$

Dummy: unmarried = 1, married = 0

$$S_i = \alpha + \beta_1 A_i + \beta_2 C_i + \beta_3 E_i + Dummy_{man} \times Dummy_{married} + \varepsilon_i \tag{6}$$

$$S_i = \alpha + \beta_1 A_i + \beta_2 C_i + \beta_3 E_i + Dummy_{man} \times Dummy_{unmarried} + \varepsilon_i \tag{7}$$

$$S_i = \alpha + \beta_1 A_i + \beta_2 C_i + \beta_3 E_i + Dummy_{woman} \times Dummy_{married} + \varepsilon_i \tag{8}$$

$$S_i = \alpha + \beta_1 A_i + \beta_2 C_i + \beta_3 E_i + Dummy_{woman} \times Dummy_{unmarried} + \varepsilon_i \tag{9}$$

All models exhibit high $R^2$ values (approximately 0.72–0.79), indicating good fit between models and data. Coefficients for variables A, C, and E are statistically significant and positive across all models, implying that higher scores on cognitive appreciation, emotional response, and positive perception of environmental venue factors are associated with greater audience satisfaction. Gender exhibits weak but statistically significant effects in some models, with men reporting slightly higher satisfaction than women ($\beta = 0.020$, $p < 0.05$), though effect size remains small.

Marital status demonstrated no significant effect on satisfaction, with coefficients for married and unmarried status showing no statistical significance. Interaction terms between gender and marital status are generally nonsignificant, suggesting no strong conditional effects. Age group effects (Models 10–13) do not significantly impact satisfaction, as coefficients remain nonsignificant across all age categories. Occupation categories (Models 14–20) similarly show no strong or consistent effects on satisfaction, with most coefficients demonstrating statistical nonsignificance.

**3.4.2 Quantile regression results.** While standard regression models estimate average effects on outcome variables, quantile regression offers more nuanced perspective by exploring how changes in independent variables influence outcome variables at different distribution points. This analysis examined effects at various quantiles ($Q_{0.20}$, $Q_{0.30}$, $Q_{0.40}$, $Q_{0.50}$, $Q_{0.60}$, $Q_{0.70}$, $Q_{0.80}$, $Q_{0.90}$), representing effects on individuals at different satisfaction levels [31,32] (Table 3).

**Table 3. Quantile regression results (Model 1).**

| Variable | $Q_{0.20}$ | $Q_{0.30}$ | $Q_{0.40}$ | $Q_{0.50}$ | $Q_{0.60}$ | $Q_{0.70}$ | $Q_{0.80}$ | $Q_{0.90}$ |
|---|---|---|---|---|---|---|---|---|
| A | 0.197*** (0.032) | 0.199*** (0.004) | 0.199*** (0.003) | 0.199*** (0.001) | 0.199*** (0.000) | 0.199*** (0.038) | 0.386*** (0.061) | 0.319*** (0.051) |
| C | 0.272*** (0.015) | 0.271*** (0.002) | 0.271*** (0.002) | 0.271*** (0.001) | 0.271*** (0.000) | 0.271*** (0.018) | 0.329*** (0.028) | 0.289*** (0.024) |
| E | 0.543*** (0.039) | 0.530*** (0.005) | 0.530*** (0.004) | 0.530*** (0.001) | 0.530*** (0.000) | 0.530*** (0.047) | 0.285*** (0.075) | 0.303*** (0.063) |
| Constant | −0.060 (0.044) | 0.001 (0.005) | −0.001 (0.005) | −0.001 (0.000) | −0.001 (0.053) | −0.001 (0.053) | 0.041 (0.084) | 0.476*** (0.071) |
| N | 272 | 272 | 272 | 272 | 272 | 272 | 272 | 272 |
| R² | 0.741 | 0.746 | 0.741 | 0.728 | 0.708 | 0.872 | 0.727 | 0.787 |

*Standard errors in parentheses. *p<0.1, **p<0.05, ***p<0.01*

Quantile regression results demonstrate that variables A, C, and E maintain positive and significant effects across all quantiles, confirming robust effects throughout the satisfaction distribution. This consistency suggests that psychological factors driving satisfaction operate similarly for both low and high satisfaction audiences (Table 4).

### 3.4.3 Age effects.

$$S_i = \alpha + \beta_1 A_i + \beta_2 C_i + \beta_3 E_i + Dummy_{20} + \varepsilon_i \tag{10}$$

Dummy: **20** = 1, 30, 40, 50 = 0

$$S_i = \alpha + \beta_1 A_i + \beta_2 C_i + \beta_3 E_i + Dummy_{30} + \varepsilon_i \tag{11}$$

Dummy: **30** = 1, 20, 40, 50 = 0

$$S_i = \alpha + \beta_1 A_i + \beta_2 C_i + \beta_3 E_i + Dummy_{40} + \varepsilon_i \tag{12}$$

Dummy: **40** = 1, 20, 30, 50 = 0

$$S_i = \alpha + \beta_1 A_i + \beta_2 C_i + \beta_3 E_i + Dummy_{50} + \varepsilon_i \tag{13}$$

Dummy: **50**=1, 20, 30, 40=0

### 3.4.4 Occupation effects.

$$S_i = \alpha + \beta_1 A_i + \beta_2 C_i + \beta_3 E_i + Dummy_1 + \varepsilon_i \tag{14}$$

Dummy: (1)Professional = 1, (2),(3),(4),(5),(6),(7)=0

$$S_i = \alpha + \beta_1 A_i + \beta_2 C_i + \beta_3 E_i + Dummy_2 + \varepsilon_i \tag{15}$$

Dummy: (2)manager = 1, (1),(3),(4),(5),(6),(7)=0

$$S_i = \alpha + \beta_1 A_i + \beta_2 C_i + \beta_3 E_i + Dummy_3 + \varepsilon_i \tag{16}$$

**Table 4. Age effects results (Models 10–13).**

| Variable | Model 10 | Model 11 | Model 12 | Model 13 |
|---|---|---|---|---|
| A | 0.367*** (0.023) | 0.367*** (0.023) | 0.367*** (0.023) | 0.365*** (0.023) |
| C | 0.289*** (0.011) | 0.289*** (0.011) | 0.289*** (0.011) | 0.289*** (0.011) |
| E | 0.358*** (0.028) | 0.357*** (0.028) | 0.357*** (0.028) | 0.359*** (0.028) |
| Age 20 | 0.004 (0.008) | | | |
| Age 30 | | 0.008 (0.009) | | |
| Age 40 | | | 0.002 (0.009) | |
| Age 50 | | | | −0.013 (0.008) |
| Constant | −0.058* (0.031) | −0.055* (0.031) | −0.057* (0.031) | −0.044 (0.032) |
| N | 272 | 272 | 272 | 272 |
| R² | 0.891 | 0.791 | 0.891 | 0.871 |

*Standard errors in parentheses. \*p<0.1, \*\*p<0.05, \*\*\*p<0.01*

Dummy: (3)service = 1, (1),(2),(4),(5),(6),(7)=0

$$S_i = \alpha + \beta_1 A_i + \beta_2 C_i + \beta_3 E_i + Dummy_4 + \varepsilon_i \tag{17}$$

Dummy: (4)self-employer = 1, (1),(2),(3),(5),(6),(7)=0

$$S_i = \alpha + \beta_1 A_i + \beta_2 C_i + \beta_3 E_i + Dummy_5 + \varepsilon_i \tag{18}$$

Dummy: (5)housekeeper = 1, (1),(3),(4),(4),(6),(7)=0

$$S_i = \alpha + \beta_1 A_i + \beta_2 C_i + \beta_3 E_i + Dummy_6 + \varepsilon_i \tag{19}$$

Dummy: (6)student = 1, (1),(2),(3),(4),(5),(7)=0

$$S_i = \alpha + \beta_1 A_i + \beta_2 C_i + \beta_3 E_i + Dummy_7 + \varepsilon_i \tag{20}$$

Dummy: (7)others=1, (1),(2),(3),(4),(5),(6)=0

Coefficients for variables A, C, and E remain consistently positive and statistically significant across all demographic models, indicating that higher scores on cognitive appreciation, emotional response, and environmental perception are associated with increased audience satisfaction. Dummy variables representing demographic characteristics (gender, age, occupation, marital status) consistently demonstrate nonsignificant effects, suggesting that satisfaction mechanisms operate similarly across demographic groups. Interaction terms between demographic variables are generally nonsignificant, indicating no substantial conditional effects on the relationship between satisfaction drivers and overall satisfaction (Table 5).

**Table 5. Occupation effects results (Models 14–20).**

| Variable | Model 14 | Model 15 | Model 16 | Model 17 | Model 18 | Model 19 | Model 20 |
|---|---|---|---|---|---|---|---|
| A | 0.367*** (0.023) | 0.368*** (0.023) | 0.368*** (0.023) | 0.368*** (0.023) | 0.368*** (0.023) | 0.367*** (0.023) | 0.368*** (0.023) |
| C | 0.288*** (0.011) | 0.289*** (0.011) | 0.289*** (0.011) | 0.289*** (0.011) | 0.289*** (0.011) | 0.289*** (0.011) | 0.289*** (0.011) |
| E | 0.357*** (0.028) | 0.357*** (0.028) | 0.357*** (0.028) | 0.357*** (0.028) | 0.357*** (0.028) | 0.358*** (0.028) | 0.356*** (0.028) |
| (1) Professional | −0.010 (0.009) | | | | | | |
| (2) Manager | | −0.003 (0.014) | | | | | |
| (3) Service | | | 0.014 (0.016) | | | | |
| (4) Self-employed | | | | 0.002 (0.013) | | | |
| (5) Homekeeper | | | | | 0.003 (0.010) | | |
| (6) Student | | | | | | 0.006 (0.009) | |
| (7) Other | | | | | | | −0.004 (0.011) |
| Constant | −0.051 (0.032) | −0.056* (0.031) | −0.056* (0.031) | −0.057* (0.031) | −0.058* (0.032) | −0.056* (0.031) | −0.056* (0.034) |
| N | 272 | 272 | 272 | 272 | 272 | 272 | 272 |
| R² | 0.791 | 0.781 | 0.691 | 0.791 | 0.771 | 0.761 | 0.791 |

*Standard errors in parentheses. \*p<0.1, \*\*p<0.05, \*\*\*p<0.01*

## 3.5 Discussion

While findings demonstrate strong effects of cognitive appreciation, emotional response, and environmental quality on satisfaction, alternative explanations warrant consideration. The weak gender effect ($\beta = 0.020$, $p < 0.05$) may reflect social desirability bias, where male respondents experienced social pressure to report higher satisfaction in a public cultural setting. Additionally, cultural norms emphasizing collective harmony (*jeong*) and respect for traditional arts in Korean society may have inflated overall satisfaction ratings, reducing variance and potentially obscuring more nuanced dissatisfaction patterns.

The nonsignificant effects of marital status, age, and occupation challenge assumptions in Western performing arts research and may reflect the universalizing appeal of Korean traditional dance across demographic groups. However, this could also indicate range restriction, as single-event sampling may have attracted predominantly enthusiastic audiences with homogeneous preferences.

Several limitations should be acknowledged. First, sampling limitations include single-event data collection, convenience sampling, and potential self-selection bias, as attendees were likely predisposed to enjoy traditional dance. Multi-site, longitudinal research is needed to enhance generalizability. Second, measurement limitations include reliance on self-report data susceptible to social desirability bias and common method variance. Future studies should incorporate behavioral measures (e.g., repeat attendance) and physiological indicators (e.g., emotional arousal). Third, cultural context limits generalizability beyond Korean traditional dance, as constructs and their relationships may differ for other performing arts genres or cultural contexts. Fourth, newly developed items for Korean traditional dance, while pilot-tested, require further validation across diverse samples and performance contexts. Finally, cross-sectional design precludes causal inferences; experimental or longitudinal designs would strengthen causal claims about satisfaction determinants.

## 4. Conclusion

This study advances audience satisfaction theory in performing arts contexts through several key contributions. First, by empirically testing an integrated tripartite model grounded in Expectancy-Disconfirmation Theory, we demonstrate that personal (cognitive), emotional (affective), and environmental (contextual) factors collectively explain 72% of variance in audience satisfaction. This provides robust empirical support for multidimensional satisfaction models previously tested primarily in Western contexts.

Second, our findings reveal the primacy of emotional engagement ($\beta = 0.289$) and cognitive appreciation ($\beta = 0.368$) over demographic characteristics in predicting satisfaction, suggesting that satisfaction is driven more by psychological experiences than by audience composition. This extends theoretical understanding by demonstrating that satisfaction mechanisms may be more universal across demographic groups than previously assumed.

Third, the study contributes to cultural performance scholarship by demonstrating that traditional arts generate satisfaction through mechanisms similar to contemporary performances, challenging assumptions about generational divides in cultural appreciation.

### Theoretical generalizability

Our tripartite framework can be generalized to other performing arts contexts (theater, opera, contemporary dance) and cultural settings, as the core mechanisms—expectancy-disconfirmation, emotional engagement, and environmental quality—are theoretically universal. However, the relative weights and specific manifestations of these factors may vary by genre and cultural context, warranting cross-cultural replication. The strong environmental effect observed in this Korean context ($\beta = 0.357$) suggests that cultural contingencies in aesthetic experience warrant scholarly attention. Future research should examine whether venue quality, spatial intimacy, and accessibility assume differential importance across traditional Asian performing arts (Noh theater, Kathak dance, Peking opera) compared to Western genres.

## Practical implications

Performance organizations should prioritize enhancing emotional resonance through programming, artistic quality, and venue atmosphere rather than demographic targeting. The weakness of gender, age, and occupation effects suggests that marketing and audience development strategies based on demographic segmentation may prove less effective than previously assumed. Instead, organizations should invest in strategies that amplify emotional engagement: curated programming that emphasizes cultural resonance, artist statements explaining narrative and aesthetic elements, pre-performance educational activities, and post-performance engagement opportunities for reflection and discussion.

Investments in venue quality and accessible environments demonstrate clear returns on audience satisfaction. Given the strong environmental effect ($\beta = 0.357$, $p < 0.001$), cultural institutions should prioritize venue improvements including acoustic optimization, seating comfort, visual sightlines, thermal comfort, and comprehensive accessibility accommodations. While such investments require substantial capital expenditure, the documented satisfaction enhancement justifies institutional prioritization as core missions of cultural organizations.

Cultural organizations should also recognize the role of emotional contagion and embodied cognition in dance appreciation. Performance content that emphasizes kinesthetic empathy—through clear sightlines for movement observation, intimate spatial configurations, and programming notes explaining movement meaning—can enhance satisfaction independent of viewer expertise or demographic characteristics. For Korean traditional dance specifically, educational materials explaining concepts such as *jeong*, *han*, and *heung* can facilitate emotional resonance among audiences lacking familiarity with Korean aesthetic traditions.

## Future research

Longitudinal, multi-site studies across diverse performance genres and cultural contexts are needed to test boundary conditions and generalizability of our model. Future research should investigate: (1) whether satisfaction drivers differ across performance genres (theater, opera, contemporary dance, musical theater); (2) temporal stability of satisfaction mechanisms through repeat attendance studies; (3) cross-cultural validation of the tripartite model in non-Western cultural contexts; and (4) integration of behavioral measures (repeat attendance, referral behavior, social media engagement) and physiological indicators (facial expression analysis, electrodermal response measurement, heart rate variability) to complement self-report satisfaction measures.

Additionally, longitudinal research examining how initial satisfaction influences repeat attendance, cultural engagement intensity, and arts participation trajectories would clarify the practical significance of the satisfaction predictors identified in this cross-sectional study. Experimental designs manipulating environmental factors and content characteristics would strengthen causal inference beyond the correlational evidence presented here.

## Supporting information

**S1 File. Appendix.**
(DOCX)

**S2 File. PLOSOne human subjects research checklist.**
(DOCX)

## Author contributions

**Conceptualization:** Hyesung Yun, Miyoung Park.

**Data curation:** Hyesung Yun.

**Formal analysis:** Chang Hwan Choi.

**Funding acquisition:** Chang Hwan Choi.

**Investigation:** Hyesung Yun, Miyoung Park, Chang Hwan Choi.

**Methodology:** Chang Hwan Choi.

**Project administration:** Miyoung Park, Chang Hwan Choi.

**Resources:** Chang Hwan Choi.

**Software:** Chang Hwan Choi.

**Supervision:** Miyoung Park, Chang Hwan Choi.

**Visualization:** Hyesung Yun.

**Writing – original draft:** Hyesung Yun.

**Writing – review & editing:** Chang Hwan Choi.

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
