## [Decision Letter · Decision Letter 0]

21 Oct 2025

Dear Dr. Choi,

Thank you for submitting your manuscript to PLOS ONE. After careful consideration, we feel that it has merit but does not fully meet PLOS ONE’s publication criteria as it currently stands. Therefore, we invite you to submit a revised version of the manuscript that addresses the points raised during the review process.

https://journals.plos.org/plosone/s/submission-guidelines#loc-laboratory-protocols . Additionally, PLOS ONE offers an option for publishing peer-reviewed Lab Protocol articles, which describe protocols hosted on protocols.io. Read more information on sharing protocols at https://plos.org/protocols?utm_medium=editorial-email&utm_source=authorletters&utm_campaign=protocols .

We look forward to receiving your revised manuscript.

Kind regards,

Zhucheng Shao

Guest Editor

PLOS ONE

**Journal Requirements:**

 Please ensure that your manuscript meets PLOS ONE's style requirements, including those for file naming. The PLOS ONE style templates can be found at https://journals.plos.org/plosone/s/file?id=wjVg/PLOSOne_formatting_sample_main_body.pdf and https://journals.plos.org/plosone/s/file?id=ba62/PLOSOne_formatting_sample_title_authors_affiliations.pdf 2. Please provide details regarding participant consent. In the ethics statement in the Methods and online submission information, please ensure that you have specified (a) whether consent was informed and (b) what type you obtained (for instance, written or verbal, and if verbal, how it was documented and witnessed). If your study included minors, state whether you obtained consent from parents or guardians. If the need for consent was waived by the ethics committee, please include this information. If you are reporting a retrospective study of medical records or archived samples, please ensure that you have discussed whether all data were fully anonymized before you accessed them and/or whether the IRB or ethics committee waived the requirement for informed consent. If patients provided informed written consent to have data from their medical records used in research, please include this information. 3. Please note that PLOS One has specific guidelines on code sharing for submissions in which author-generated code underpins the findings in the manuscript. In these cases, we expect all author-generated code to be made available without restrictions upon publication of the work. Please review our guidelines at https://journals.plos.org/plosone/s/materials-and-software-sharing#loc-sharing-code and ensure that your code is shared in a way that follows best practice and facilitates reproducibility and reuse. 4. When completing the data availability statement of the submission form, you indicated that you will make your data available on acceptance. We strongly recommend all authors decide on a data sharing plan before acceptance, as the process can be lengthy and hold up publication timelines. Please note that, though access restrictions are acceptable now, your entire data will need to be made freely accessible if your manuscript is accepted for publication. This policy applies to all data except where public deposition would breach compliance with the protocol approved by your research ethics board. If you are unable to adhere to our open data policy, please kindly revise your statement to explain your reasoning and we will seek the editor's input on an exemption. Please be assured that, once you have provided your new statement, the assessment of your exemption will not hold up the peer review process. 5. Please amend either the abstract on the online submission form (via Edit Submission) or the abstract in the manuscript so that they are identical. 6. Please include your full ethics statement in the ‘Methods’ section of your manuscript file. In your statement, please include the full name of the IRB or ethics committee who approved or waived your study, as well as whether or not you obtained informed written or verbal consent. If consent was waived for your study, please include this information in your statement as well. 7. If the reviewer comments include a recommendation to cite specific previously published works, please review and evaluate these publications to determine whether they are relevant and should be cited. There is no requirement to cite these works unless the editor has indicated otherwise. 

**Additional Editor Comments:**

We are writing to inform you of the latest update for your manuscript. We have now received feedback from experienced reviewers. Based on their comments, we decided to make a Minor Revision for the manuscript at this stage. You can find the comments from reviewers below, and we look forward to your revised submission.

The research questions should be clearly articulated in the introduction section.A more critical review of the existing literature is needed to identify the knowledge gaps, which should be explicitly linked to the research objectives.More detailed information on the data collection process would strengthen the study’s transparency and replicability.The discussion section could benefit from a broader comparison with previous research findings.

Reviewers' comments:

**Comments to the Author**

1. Is the manuscript technically sound, and do the data support the conclusions?

Reviewer #1: Yes

2. Has the statistical analysis been performed appropriately and rigorously?

Reviewer #1: Yes

3. Have the authors made all data underlying the findings in their manuscript fully available?

Reviewer #1: No

4. Is the manuscript presented in an intelligible fashion and written in standard English?

Reviewer #1: Yes

**Reviewer #1: ** This report provides a critical review of the manuscript entitled “Behavioral Insights into Audience Satisfaction: Analyzing Emotional and Cognitive Factors in K-Culture Dance Engagement.” The review identifies strengths, weaknesses, and areas for improvement.

Comments

1. Title/Abstract: The title is specific. The abstract is however descriptive and does not contain explicit statement of theoretical contribution. The authors ought to put emphasis on the contribution of the study to theory and practice in the study of audience satisfaction.

2. Introduction: The statement of the problem is to be expressed in a clearer manner and then be succeeded by a number of research questions or hypotheses.

3. Literature Review: The literature review is extensive and informative but in some cases it reminds of a cultural essay more than a specialized synthesis. Certain areas (e.g. digital engagement, AR/Tik Tok) are not obviously aligned with the empirical models. The review needs to be minimal in terms of supporting the variables of the study.

4. Theoretical Framework: Despite the mentioning of the factors, emotional, cognitive and environmental, the paper lacks a clearly defined anchoring theory. It should provide a conceptual model which is connected to a particular framework like Expectancy-Disconfirmation or Social Cognitive Theory.

5. Methodology: Data collection is outlined in detail but the sampling method used (one performance event) constrains representativeness. Also, there is no clarity in the validation of newly developed items. Pilot testing, reliability and validity tests should be reported in the paper.

6. Discussion: The discussion paraphrases results, but does not critically reflect. Other reasons like culture norm or social desirability bias ought to be addressed. There is also no clear limitations section present.

7. Conclusion: The conclusion cannot be described as adequate focus on theoretical progress. The authors ought to explain the ways in which their results can be generalized to the audience satisfaction theory.

8. References and Formatting: The reference list is very long however it contains secondary references, repeated references and broken links. The authors are to give the preference to peer-reviewed sources and adhere to APA formatting.

**Do you want your identity to be public for this peer review?**  For information about this choice, including consent withdrawal, please see our Privacy Policy

Reviewer #1: **Yes: ** Dr. Ayesha Noreen

---

## [Author Response · Author response to Decision Letter 1]

27 Oct 2025

Response to the Review

Dear Editor,

Thank you for the opportunity to revise our manuscript titled "Behavioral Insights into Audience Satisfaction: Analyzing Emotional and Cognitive Factors in K-Culture Dance Engagement" for consideration in PLOS ONE. We sincerely appreciate the time and effort that you, Reviewer #1, and the editorial team have dedicated to providing valuable feedback on our manuscript. We are grateful for the insightful comments that have significantly strengthened the quality of our work.

We have carefully addressed all the concerns raised by the reviewer and have incorporated the suggested changes throughout the manuscript. Below, we provide a detailed point-by-point response to each comment, indicating the specific revisions made and their locations in the revised manuscript (marked in red text). We have also addressed all journal requirements outlined in your decision letter.

Comment 1: Title/Abstract

The title is specific. The abstract is however descriptive and does not contain explicit statement of theoretical contribution. The authors ought to put emphasis on the contribution of the study to theory and practice in the study of audience satisfaction.

Response: We agree with this important observation. We have revised the abstract to explicitly articulate our theoretical contributions. The revised abstract now emphasizes how our study advances audience satisfaction theory by integrating personal, emotional, and environmental factors within the Korean traditional dance context, and demonstrates practical implications for cultural practitioners.

Revision (Abstract, Page 1, Lines 8-12): Added the following statement:

"This research advances audience satisfaction theory by demonstrating that the tripartite model of personal orientation, emotional engagement, and environmental quality collectively explains over 70% of variance in satisfaction levels. These findings provide both theoretical insights into cultural performance appreciation and practical guidance for enhancing audience experiences in traditional performing arts contexts."

Comment 2: Introduction

The statement of the problem is to be expressed in a clearer manner and then be succeeded by a number of research questions or hypotheses.

Response: Thank you for this valuable feedback. We have substantially revised the introduction to provide a clearer problem statement and have now included explicit research questions that guide our empirical investigation.

Revision (Introduction, Page 3, Lines 15-28): Added clear problem statement:

"Despite the growing global interest in Korean traditional dance and its increasing integration with digital technologies, empirical research examining the multidimensional factors influencing audience satisfaction remains limited. While existing studies have explored individual aspects such as emotional responses or venue characteristics, a comprehensive framework integrating personal, emotional, cognitive, and environmental factors is lacking."

Research Questions Added (Introduction, Page 4, Lines 1-8):

• RQ1: To what extent do personal orientation factors (personality characteristics) influence audience satisfaction with Korean traditional dance performances?

• RQ2: How do emotional and cognitive responses affect overall audience satisfaction?

• RQ3: What role do environmental factors (venue quality, accessibility) play in shaping satisfaction levels?

• RQ4: Are there significant demographic differences (gender, age, marital status, occupation) in audience satisfaction patterns?

Comment 3: Literature Review

The literature review is extensive and informative but in some cases it reminds of a cultural essay more than a specialized synthesis. Certain areas (e.g. digital engagement, AR/TikTok) are not obviously aligned with the empirical models. The review needs to be minimal in terms of supporting the variables of the study.

Response: We appreciate this constructive criticism. We have streamlined the literature review to focus

specifically on studies that directly support our research variables (personal orientation, emotional/cognitive responses, and environmental factors). We have removed tangential discussions about digital platforms, AR, and TikTok that are not directly measured in our empirical model, and reorganized the review to align closely with our research framework.

Revisions (Literature Review, Pages 5-12):

• Removed sections 2.1.2 and 2.3.2 discussing AR/TikTok platforms (approximately 800 words deleted)

• Reorganized sections to mirror our empirical variables: Section 2.1 (Personal Characteristics and Satisfaction), Section 2.2 (Emotional and Cognitive Processing), Section 2.3 (Environmental Factors), Section 2.4 (Research Gap)

• Added explicit connections between reviewed literature and our measured constructs throughout

• Reduced literature review length by approximately 30% while strengthening theoretical alignment

Comment 4: Theoretical Framework

Despite the mentioning of the factors, emotional, cognitive and environmental, the paper lacks a clearly defined anchoring theory. It should provide a conceptual model which is connected to a particular framework like Expectancy-Disconfirmation or Social Cognitive Theory.

Response: This is an excellent point that significantly strengthens our theoretical contribution. We have added a comprehensive theoretical framework section that explicitly grounds our study in established theories.

Revision (New Section 2.5: Theoretical Framework, Pages 12-13, Lines 5-35):

"This study is grounded in the Expectancy-Disconfirmation Theory (EDT) [Oliver, 1997] and the Tripartite Model of Attitude [Rosenberg & Hovland, 1960], which together provide a comprehensive framework for understanding audience satisfaction.

Expectancy-Disconfirmation Theory posits that satisfaction results from the comparison between pre-performance expectations and actual performance experiences. When experiences exceed expectations (positive disconfirmation), satisfaction increases; when experiences fall short (negative disconfirmation), satisfaction decreases [Oliver, 1997]. In our context, audience members form expectations about Korean traditional dance based on personal characteristics and prior exposure, and these expectations interact with actual performance experiences to determine satisfaction.

The Tripartite Model of Attitude suggests that attitudes (including satisfaction) are formed through three components: cognitive (beliefs and thoughts), affective (emotions and feelings), and behavioral (past experiences and actions) [Rosenberg & Hovland, 1960]. Our empirical model operationalizes these components as:

• Cognitive component (A): Personal appreciation and rational evaluation of dance quality

• Affective component (C): Emotional responses and engagement during performance

• Behavioral/contextual component (E): Environmental factors influencing the performance experience

Conceptual Model: Figure 1 (Page 13) presents our integrated conceptual framework, illustrating how personal characteristics, emotional/cognitive responses, and environmental factors collectively influence audience satisfaction through the mechanisms proposed by EDT and the Tripartite Model."

[Figure 1 added: Conceptual framework diagram showing theoretical pathways]

Comment 5: Methodology - Data Collection

Data collection is outlined in detail but the sampling method used (one performance event) constrains representativeness. Also, there is no clarity in the validation of newly developed items. Pilot testing, reliability and validity tests should be reported in the paper.

Response: We acknowledge this important methodological concern and have substantially expanded the methodology section to address these limitations and provide comprehensive validation information.

Revision (Section 3.1: Sample and Procedure, Pages 14-15):

Sampling Limitations Acknowledged (Page 14, Lines 20-28): "We acknowledge that data collection from a single performance event on November 25, 2023, may limit the generalizability of findings across different performance contexts, venues, and time periods. This convenience sampling approach was necessitated by practical constraints but represents a limitation that should be considered when interpreting results. Future research should employ multi-site, multi-performance sampling to enhance external validity."

Revision (Section 3.1.1: Instrument Development and Validation, Pages 15-16, NEW):

"Pilot Testing: Prior to the main data collection, a pilot study (N=45) was conducted with audiences at a preliminary performance in October 2023. Pilot participants completed the questionnaire and provided feedback on item clarity, comprehension, and appropriateness. Based on pilot results, three items were reworded for clarity, and one redundant item was removed.

Scale Development: The questionnaire comprised 20 items across four constructs: (1) Personal Orientation (6 items), (2) Emotional and Cognitive Response (5 items), (3) Environmental Factors (5 items), and (4) Overall Satisfaction (4 items). Items were measured on a 5-point Likert scale (1=Strongly Disagree, 5=Strongly Agree). Most items were adapted from validated instruments in performing arts research [Hume & Mort, 2010; Radbourne et al., 2010]; newly developed items specific to Korean traditional dance were created following established scale development procedures [DeVellis, 2016].

Reliability: Internal consistency was assessed using Cronbach's alpha coefficients:

• Personal Orientation: α = 0.78

• Emotional/Cognitive Response: α = 0.81

• Environmental Factors: α = 0.86

• Overall Satisfaction: α = 0.89

All scales exceeded the recommended threshold of 0.70 [Nunnally & Bernstein, 1994], indicating acceptable to excellent reliability.

Validity: Construct validity was assessed through Principal Component Analysis (PCA) with varimax rotation (see Appendix A). Kaiser-Meyer-Olkin (KMO) values ranged from 0.625 to 0.857, exceeding the 0.60 threshold, and Bartlett's Test of Sphericity was significant (p<0.001) for all scales, confirming factorability. Convergent validity was demonstrated by significant correlations among related constructs (r = 0.45 to 0.67, all p<0.01). Discriminant validity was supported by factor loadings showing clear separation of constructs in PCA."

Comment 6: Discussion

The discussion paraphrases results, but does not critically reflect. Other reasons like cultural norms or social desirability bias ought to be addressed. There is also no clear limitations section present.

Response: We agree that the discussion requires more critical depth. We have substantially revised this section to include critical reflection, alternative explanations, and potential biases.

Revision (Section 4: Discussion, Pages 22-25, SUBSTANTIALLY REVISED):

Added Critical Reflection (Page 22, Lines 15-30):

"While our findings demonstrate strong effects of appreciation, emotional response, and venue quality on satisfaction, alternative explanations warrant consideration. The weak gender effect (β=0.02, p<0.05) may reflect social desirability bias, where male respondents felt social pressure to report higher satisfaction in a public cultural setting. Additionally, cultural norms emphasizing collective harmony (jeong) and respect for traditional arts in Korean society may have inflated overall satisfaction ratings, reducing variance and potentially obscuring more nuanced dissatisfaction patterns [Kim & Kim, 2018].

The non-significant effects of marital status, age, and occupation challenge assumptions in Western performing arts research [Brown & Miller, 2018] and may reflect the universalizing appeal of Korean traditional dance across demographic groups. However, this could also indicate range restriction, as our single-event sampling may have attracted predominantly enthusiastic audiences with homogeneous preferences."

Added Section 4.5: Limitations (Page 24, NEW):

"4.5 Limitations

Several limitations should be acknowledged. First, sampling limitations include single-event data collection, convenience sampling, and potential self-selection bias, as attendees were likely predisposed to enjoy traditional dance. Multi-site, longitudinal research is needed to enhance generalizability.

Second, measurement limitations include reliance on self-report data susceptible to social desirability bias and common method variance. Future studies should incorporate behavioral measures (e.g., repeat attendance) and physiological indicators (e.g., emotional arousal).

Third, cultural context limits generalizability beyond Korean traditional dance. The constructs and their relationships may differ for other performing arts genres or cultural contexts.

Fourth, newly developed items for Korean traditional dance, while pilot-tested, require further validation across diverse samples and performance contexts.

Finally, cross-sectional design precludes causal inferences. Experimental or longitudinal designs would strengthen causal claims about satisfaction determinants."

Comment 7: Conclusion

The conclusion cannot be described as adequate focus on theoretical progress. The authors ought to explain the ways in which their results can be generalized to the audience satisfaction theory.

Response: We have substantially revised the conclusion to emphasize theoretical contributions and generalizability to broader audience satisfaction theory.

Revision (Section 5: Conclusion, Page 26, Lines 1-30, SUBSTANTIALLY REVISED):

"5. Conclusion

This study advances audience satisfaction theory in performing arts contexts through several key contributions. First, by empirically testing an integrated tripartite model grounded in Expectancy-Disconfirmation Theory, we demonstrate that personal (cognitive), emotional (affective), and environmental (contextual) factors collectively explain 72% of variance in audience satisfaction. This provides robust empirical support for multidimensional satisfaction models previously tested primarily in Western contexts [Hume & Mort, 2010].

Second, our findings reveal the primacy of emotional engagement (β=0.289) and cognitive appreciation (β=0.368) over demographic characteristics in predicting satisfaction, suggesting that satisfaction is driven more by psychological experiences than by audience composition. This extends theoretical understanding by demonstrating that satisfaction mechanisms may be more universal across demographic groups than previously assumed [Radbourne et al., 2010].

Third, the study contributes to cultural performance scholarship by demonstrating that traditional arts generate satisfaction through mechanisms similar to contemporary performances, challenging assumptions about generational divides in cultural appreciation.

Theoretical Generalizability: Our tripartite framework can be generalized to other performing arts contexts (theater, opera, dance) and cultural settings, as the core mechanisms—expectancy-disconfirmation, emotional engagement, and environmental quality—are theoretically universal. However, the relative weights and specific manifestations of these factors may vary by genre and cultural context, warranting cross-cultural replication.

Practical Implications: Performance organizations should prioritize enhancing emotional resonance through programming, artistic quality, and venue atmosphere rather than demographic targeting. Investments in venue quality and accessible environments demonstrate clear returns on audience satisfaction.

Future Research: Longitudinal, multi-site studies across diverse performance genres and cultural contexts are needed to test boundary conditions and generalizability of our model."

Comment 8: References and Formatting

The reference list is very long however it contains secondary references, repeated references and broken links. The authors are to give the preference to peer-reviewed sources and adhere to APA formatting.

Response: We have thoroughly audited and revised the reference list to comply with APA 7th edition and PLOS ONE Vancouver style requirements.

Anderson, J. (1992). Art as communication: Redesigning the aesthetics of dance. Routledge.

Armstrong, J. S., & Overton, T. S. (1977

---

## [Decision Letter · Decision Letter 1]

10 Nov 2025

Behavioral Insights into Audience Satisfaction: Analyzing Emotional and Cognitive Factors in K-Culture Dance Engagement

PONE-D-25-39772R1

Dear Dr. Choi,

We’re pleased to inform you that your manuscript has been judged scientifically suitable for publication and will be formally accepted for publication once it meets all outstanding technical requirements.

Kind regards,

Zhucheng Shao

Guest Editor

PLOS ONE

Reviewer's Responses to Questions

**Comments to the Author**

Reviewer #1: All comments have been addressed

2. Is the manuscript technically sound, and do the data support the conclusions?

Reviewer #1: Yes

3. Has the statistical analysis been performed appropriately and rigorously?

Reviewer #1: Yes

4. Have the authors made all data underlying the findings in their manuscript fully available?

Reviewer #1: Yes

5. Is the manuscript presented in an intelligible fashion and written in standard English?

Reviewer #1: Yes

Reviewer #1: Overall, all major reviewer concerns have been satisfactorily addressed. The revised manuscript demonstrates clear theoretical grounding, enhanced methodological rigor, and improved academic quality compared to the previous version. The authors have effectively incorporated the earlier feedback by refining the abstract, strengthening the theoretical framework, improving methodological transparency, and ensuring better alignment between analysis, discussion, and conclusion.

Below is the detailed analysis of comments in relation to the previous and revised versions.

1. Abstract:

Previously, the abstract lacked theoretical contribution.

The author revised it to include explicit theoretical and practical implications using Expectancy-Disconfirmation Theory and the Tripartite Model.

2. Introduction:

Earlier, the problem statement and research questions were unclear.

The revised version includes a clear problem statement and four specific research questions aligned with the study objectives.

3. Literature Review/Theoretical Framework:

Previously, the review was too descriptive and lacked a clear theoretical base.

The author streamlined it, removed irrelevant parts (e.g., AR/TikTok), and added a solid theoretical framework with a conceptual model.

4. Methodology:

Earlier, there was no clarity on item validation or sampling limitations.

The revised paper includes details on pilot testing, reliability/validity tests, and acknowledges single-event sampling as a limitation.

5. Discussion and Conclusion:

Previously, lacked critical reflection, limitations, and theoretical focus.

The author added discussion on cultural bias, included a full limitations section, and strengthened the conclusion with theoretical generalization.

**Do you want your identity to be public for this peer review?** For information about this choice, including consent withdrawal, please see our Privacy Policy

Reviewer #1: **Yes: ** Ayesha Noreen

---

## [Editor Report · Acceptance letter]

PONE-D-25-39772R1

PLOS ONE

Dear Dr. Choi,

I'm pleased to inform you that your manuscript has been deemed suitable for publication in PLOS ONE. Congratulations! Your manuscript is now being handed over to our production team.

Kind regards,

on behalf of

Dr. Zhucheng Shao

Guest Editor

PLOS ONE